# Chronic Nonbacterial Osteomyelitis and Inflammatory Bowel Disease: A Literature Review-Based Cohort

**DOI:** 10.3390/children10030502

**Published:** 2023-03-03

**Authors:** Stefania Costi, Sabino Germinario, Marco Pandolfi, Maria Rosa Pellico, Andrea Amati, Maurizio Gattinara, Cecilia Beatrice Chighizola, Roberto Caporali, Achille Marino

**Affiliations:** 1Unit of Pediatric Rheumatology, ASST G. Pini-CTO, 20122 Milan, Italy; 2University of Milan, 20122 Milan, Italy; 3Department of Clinical Sciences and Community Health, Research Center for Pediatric and Adult Rheumatic Diseases (RECAP.RD), University of Milan, 20122 Milan, Italy; 4Department of Rheumatology and Medical Sciences, ASST G. Pini-CTO, 20122 Milan, Italy

**Keywords:** chronic nonbacterial osteomyelitis, inflammatory bowel diseases, bowel inflammation, bone inflammation

## Abstract

Background: Chronic nonbacterial osteomyelitis (CNO) is a rare autoinflammatory bone disorder that mainly involves children and adolescents. The association with other inflammatory disorders, such as inflammatory bowel disease (IBD), psoriasis, and arthritis, has been reported in the literature. In particular, the relationship between bone and intestinal inflammation is still poorly understood. For this purpose, our review aims to describe the cases reported in the literature concerning this association and to compare them with data from our single-center cohort of patients. Methods: We conducted a literature review of published cases of CNO associated with IBD. Eligible articles were identified through a Medline search in the PubMed database until December 2022. We retrospectively reviewed medical records of patients with CNO referred to G. Pini Hospital and compared them with the literature-review-based cohort. Results: Fifty-seven patients with a defined diagnosis of CNO and associated IBD were described in the literature (female 55%). The median age of onset of the disease (CNO or IBD) was 11 years. In 32/53 (60%), a diagnosis of Crohn’s disease (CD) was made, while 18 (34%) patients were classified as suffering from ulcerative colitis (UC) and 3 (6%) from undifferentiated IBD. The diagnosis of CNO preceded the diagnosis of IBD in 59% of cases; while in 24%, IBD anticipated CNO; and in 17%, the two conditions appeared simultaneously. The median time between the two events was 24 months. In our Italian cohort (*n* = 23 patients), no diagnosis of IBD was made. No significant differences were found when comparing clinical and demographical characteristics of the Italian vs. review-based cohort, except for a significant involvement of rachis in the Italian group. Conclusions: The correlation between autoinflammatory bone disease and intestinal inflammation should be further investigated. It is essential to promote awareness among pediatric rheumatologists and gastroenterologists about this possible association to facilitate the diagnosis and better optimize treatment.

## 1. Introduction

### 1.1. Chronic Nonbacterial Osteomyelitis: From Pathogenesis to Clinical Aspects

Chronic nonbacterial osteomyelitis (CNO) is a rare autoinflammatory disease of bone primarily affecting children and adolescents [1,2]. CNO includes various forms of bone inflammation, from mild and self-limiting conditions to chronic and severe multifocal diseases such as chronic recurrent multifocal osteomyelitis (CRMO).

CNO pathogenesis is largely unknown; however, an aberrant regulation of the innate immune system leading to cytokines disproportion toward a pro-inflammatory state (high level of IL-1, IL-6, and TNF-α and low level of IL-10) seems to promote the recurrence of bone inflammation [3]. In particular, it has been demonstrated that a fixed expression of pro-inflammatory cytokines and chemokines, promoted by the activation of the NLR family pyrin domain containing 3 (NLRP3) inflammasome [4], regulates osteoclast differentiation and activation through interactions between the receptor activator of nuclear factor-κB (RANK) on osteoclast precursor cells and its ligand (RANKL) [5,6]. NLRP3 is a central component of the innate immune system and, through caspase-1 activation, promotes the conversion to the active form of IL-1β. In support of this argument, an increased expression of NLRP3 inflammasome and IL-1β mRNA in peripheral blood mononuclear cells from CNO/CRMO patients has been described [3,7]. Indeed, Hofmann et al. reported reduced IL-10 and IL-19 expression contributing to NLRP3 inflammasome activation [3].

The majority of CNO patients do not show a clear genetic background, although, susceptible loci have been identified [8], suggesting a possible role of genetic factors in the pathogenesis of CNO/CRMO. There is evidence of family clusters of CNO/CRMO, and it has been shown that almost 50% of people diagnosed with CNO also have a personal or family history of other autoimmune or inflammatory diseases [9,10,11,12,13]. Genetic factors’ importance is also supported by the strong similarities between the sporadic form of CNO/CRMO and the monogenic ones. The most known monogenic forms of CNO/CRMO are Majeed Syndrome and Interleukin-1 Receptor Antagonist Deficiency Syndrome (DIRA). Both conditions are classified as autoinflammatory diseases, and belong to the group of ‘inflammasomopathies’, in which different mutations in genes encoding for inflammasome components or inflammasome regulator proteins lead to a pathological increase in IL-1β levels [14,15]. It is inherited in an autosomal recessive pattern. The syndrome is characterized by an early onset of sterile bone inflammation, dyserythropoietic anemia, and neutrophilic dermatosis. Disease onset is within 2 years of age and is usually associated with fever attacks, growth disturbances, and joint contractures. Some Majeed patients do not show skin manifestations. The efficacy of IL-1 blocking agents has indirectly confirmed the pathogenesis of Majeed syndrome [16,17]. DIRA results from a loss of-function mutations in IL1RN, causing a decrease in IL-1 receptor antagonist (IL-1RA) which promotes pro-inflammatory IL-1 signaling. Patients with DIRA show both skin and bone involvement along with systemic inflammation.

Indeed, it is characterized by multifocal sterile osteitis, periostitis, and pustulosis. Pustular lesions appear within the first months of life, while osteitis and periostitis occur later. Bone lesions show neutrophilic infiltration at biopsies. Widening of the anterior rib ends and periostitis of long bones are classically described in these patients. The exogenous IL-1 receptor antagonist, anakinra, is able to neutralize both IL-1α and IL-1β and is successfully employed in DIRA patients [18,19]. Deficiency of the interleukin (IL)-36 receptor antagonist (DITRA) is a recently described autosomal recessive autoinflammatory syndrome characterized clinically by recurrent episodes of generalized pustular psoriasis [20]. Cherubism is another rare monogenic autoinflammatory bone disease due to heterozygous mutations of SH3 binding protein 2 (SH3BP2) gene with variable penetrance and phenotypic manifestation within the same family. SH3BP2 plays a key role in osteoclast activation fostering bone resorption [21]. These patients show osteolytic lesions of mandibles very early in life; the following fibrotic tissue deposition and bone remodeling give the typical aspect of puffy cheeks to patients with cherubism. Cherubism treatment is based on case reports in which adalimumab and bisphosphonates have both been used. Cox et al. described a South Asian child, born to consanguineous parents, with CRMO and psoriasis carrying the homozygous mutation of the FBLIM1 gene [22]. The FBLIM1 gene encodes for Filamin-binding LIM protein 1 (FBLP1) or migfilin; this protein plays a key role in bone remodeling. FBLP1 regulates cytoskeleton activity; it binds adhesion proteins and filamin-containing actin filaments of cell-extracellular matrix (ECM) and participates in integrin activation. Mice deficient in FBLP1 show severe osteopenia due to increased osteoclast activation fostered by RANKL overexpression [23]. The FBLIM1 role in sterile bone inflammation is confirmed by a mouse model (Pstpip2-deficient) with chronic multifocal osteomyelitis that showed levels of FBLIM1 gene expression downregulated up to 20-fold compared to healthy mice [22]. After this proband, Cox et al., found another subject harboring a compound heterozygosis of FBLIM1 gene among 96 CRMO patients screened. More recently, an Italian group found a high frequency (22%) of rare variants of FBLIM1 in a multicentric cohort of patients with CNO. These data support the role of FBLIM1 in the pathogenesis of CNO [24].

CNO patients typically have a disease-onset age of around 10 years, with girls more frequently affected (female/male ratio of approximately 2:1) [9,11,25,26,27]. Although CNO has been considered to be quite rare, recent observations documented an increased incidence in tertiary centers [10,28]. CNO is probably underestimated due to diagnostic difficulties and low awareness among clinicians. Indeed, CNO is characterized by an insidious onset and variable clinical manifestations that may mimic infections or malignancies, and investigations (laboratory features and imaging) are non-specific [29]. The most common symptom at onset is insidious bone pain, and physical examinations may show tenderness to the affected site with or without local swelling [29], even though asymptomatic bone inflammation sites are frequently detected through imaging. Constitutional symptoms (low-grade fever, slight malaise, and fatigue) occur in less than 5% of patients [9,30]. Nocturnal bone pain may be present in some patients and can be misinterpreted as growing pains [13]. Bone lesions are usually multifocal and symmetric; long bones are frequently involved. Conventional radiography can identify osteolytic lesions with sclerosis (typical of an advanced stage of disease), but it is less useful at disease onset. On the other hand, magnetic resonance imaging (MRI), with short tau inversion recovery (STIR) sequences, is a more sensitive technique that can detect early bone lesions [31]. In this setting, whole-body MRI may help discriminate between mono or multifocal disease and is extremely useful in disease follow-up. Nowadays, CNO/CRMO remains a diagnosis of exclusion, and bone biopsies may allow for the clinician to rule out other diseases, such as tumors or infective osteomyelitis. The most common histological findings from CNO patients include dense infiltrates of immune cells, belonging both to the innate and adaptive immune systems, osteolysis, and fibrosis [32].

### 1.2. Inflammatory Bowel Diseases in Children: From Pathogenesis to Clinical Aspects

Inflammatory bowel diseases (IBDs) are a group of diseases characterized by immune activation and inflammation of the gastrointestinal tract. IBDs include Crohn’s disease (CD), ulcerative colitis (UC), and undifferentiated IBD (u-IBD) [33]. The pathogenesis of the disease is not yet clearly understood, but genetic factors, environmental insults, and dysbiosis seem to cooperate to determine the disease [34]. In spite of different phenotypes, approximately 30% of IBD-related genes are shared between CD and UC, suggesting common pathogenetic pathways; indeed, more than 50% of IBD susceptibility loci have also been related to other inflammatory/autoimmune diseases [34]. NOD2, C13orf31, and LRRK2 genes are associated both with susceptibility to CD and *Mycobacterium leprae* infection. Interestingly, NOD2 mutation has been shown to contribute to the dysfunction in the mucosal barrier integrity in CD. This has further led to an intense investigation into the impact of the gut microbiome dysbiosis in IBD.

In patients with a very early onset of IBD (before the age of 6), genetics acquires a priority role. In particular, IL-10 signaling pathway defects are associated with the development of early-onset IBD [35]. In contrast, IL23R allele variants have shown a protective role. In fact, T helper 17 lymphocytes from IBD patients (carriers of protective IL23R variant) show a decreased production in IL-17A in response to IL-23 stimulation [36,37]. In 20–30% of cases, IBDs are diagnosed in childhood or in young adulthood with male predominance [38]. The incidence of the disease is increasing in industrialized countries, suggesting that environmental and dietary factors play a key role [39,40]. In children, a positive family history of IBD remains an important risk factor for the development of IBD [41].

The most common symptoms and signs are diarrhea, weight loss, and abdominal pain. In UC, rectal bleeding is present in 83% of pediatric patients [42,43]. Fecal calprotectin, intestinal ultrasound, and magnetic resonance enterography may help with diagnosis, but esophagogastroduodenoscopy and colonoscopy remain the gold standard for diagnosis and classification. Extra-intestinal manifestations (EIM) can affect musculoskeletal (arthritis, osteopenia), dermatologic (pyoderma granulosum, erythema nodosum), hepatobiliary (pancreatitis, primary sclerosing cholangitis), renal (uveitis, episcleritis, iritis), and hematologic (anemia, deep venous thromboembolism) systems. Some EIM go hand in hand with intestinal inflammation, while others are independent (i.e., pyoderma gangrenosum) [33]. Joint involvement represents the most common EIM in IBD; commonly, it occurs within the first year after the onset of IBD symptoms. However, arthritis might be the presenting symptom of IBD in up to 4% of pediatric IBD cases [44]. The pattern of joint involvement might be heterogeneous: asymmetrical polyarticular or oligoarticular arthritis, arthralgia, enthesitis, and sacroiliitis [44]. During the disease course of pediatric IBD, sacroiliitis might be underrecognized due to the subtle symptoms, and can progress despite the IBD treatments. MR enterography (MRE) is used to assess IBD intestinal disease activity and might help to detect the underlining inflammation of sacroiliac joints. Indeed, a reevaluation of MRE in a cohort of children with IBD showed that, using the ASAS (Assessment of SpondyloArthritis international Society) criteria for signs of sacroiliitis, 15% of patients have mild sacroiliitis at MRE despite not complaining of any musculoskeletal symptoms [45]. The link between IBD and joint/bone inflammation might rely on the alteration of IL-23 and/or IL-17 axis, dysbiosis, and subclinical gut barrier dysfunction as well as genetic factors such as the HLA B27 positivity and NOD2 variants [46,47].

### 1.3. Association between CNO/CRMO and IBD

Since the first descriptions, the association with other inflammatory disorders such as inflammatory bowel disease (IBD), psoriasis, and arthritis has been reported [26,48]. In particular, the relationship between bone and intestinal inflammation is still poorly understood. On the other hand, the prevalence of musculoskeletal involvement in IBD is not negligible (up to 46%) [49], but the impact of CNO is unknown. In recent years, there has been a growing interest in the role of the gut microbiome in CNO. Lukens et al. demonstrated that a high-fat diet reduced Prevotella levels and pro-IL-1-beta expression in distant neutrophils in osteomyelitis-prone cmo mice [50]. In another study, an alteration in oral microbiome in CNO/CRMO patients in response to NSAID treatment was described. Another study described an alteration in the oral microbiome in CNO/CRMO patients responding to NSAID treatment [51]. A recent study has also highlighted compositional differences in fecal and oral microbiota between healthy controls and CRMO patients, drawing new attention to the potential role of dysbiosis in the pathology of CRMO [52].

This review aims to describe all the cases reported in the literature concerning the association between CNO and IBD and to compare them with data from our single-center cohort of patients affected by CNO but without a diagnosis of IBD. The comparison was assessed to comprehend if there are clinical or demographic differences between CNO alone or in association with IBD.

## 2. Materials and Methods

We conducted a literature review of published cases of CNO associated with IBD. Only articles published in peer-reviewed journals were included. Eligible articles were identified through a Medline search in the PubMed database until December 2022. No age limits or language restrictions were applied. To identify relevant reports, we used “Chronic nonbacterial osteomyelitis” OR “chronic recurrent multifocal osteomyelitis“ AND “inflammatory bowel diseases” OR “Crohn’s disease” OR “Ulcerative colitis” as search terms. Selected articles were manually screened by reading the titles and abstracts, and additional articles referenced by the above studies were reviewed to determine eligibility. We extrapolated individual patients’ data to perform a detailed analysis, creating a customized database. S.C. and S.G. reviewed the completed database to verify possible abstracting mistakes and/or case overlaps. Information not clearly stated in the text was labeled as unavailable. We retrospectively reviewed medical records of patients with CNO referred to G. Pini Hospital, and we compared demographic and clinical features from the literature-review-based cohort with our single-center cohort of patients affected by CNO. Descriptive statistics were used to summarize data. Categorical variables were expressed as percentages. The normality of variable distribution was tested by applying the Shapiro–Wilk test. Continuous variables with non-parametric distribution were expressed as median values (interquartile range, IQR). Associations between categorical predictors and outcomes were evaluated by Chi-square or Fisher’s exact test, as appropriate. *p*-values ≤ 0.05 were considered statistically significant. Analyses were performed with R commander.

## 3. Results

### 3.1. Literature-Review-Based Cohort

We identified 32 articles, 4 of which were excluded since they lacked essential data or the article was unavailable. The remaining 28 were considered eligible for the study [11,53,54,55,56,57,58,59,60,61,62,63,64,65,66,67,68,69,70,71,72,73,74,75]. Fifty-seven patients with a defined diagnosis of CNO and associated IBD were described (female 26/47; 55%) through a literature review of published cases. The median age of onset of the disease (IBD or CNO) was 11 years (IQR 4.5); in particular, eight patients were adults at onset, while in the remaining patients, the condition had started before the age of 18. Among the patients, 35/42 (83%) had a polyostotic disease, while 7/42 (17%) had a mono-focal disease with a predominance of sternoclavicular involvement (3/7; 43%). The most frequent bone symptom was wrist pain (24%), followed by nonspecific pain in the lower extremities (19%). Constitutional symptoms were present in 21% of patients at the onset of the disease (*n* = 5 with a first diagnosis of CNO; *n* = 1 first diagnosis of IBD; *n* = 3 simultaneous diagnosis). Clinical features of the retrieved cases are summarized in Table 1.

The most commonly associated conditions were psoriasis (*n* = 5; 9%), sclerosing cholangitis (*n* = 2; 4%), and erythema nodosum (*n* = 2; 4%), while the other three patients suffered from Takayasu vasculitis, pyoderma gangrenosum, and Ehlers–Danlos syndrome. In 32/53 patients (60%), a histological diagnosis of CD was made, while 18 (34%) patients were classified as UC. In three (6%) cases, the bowel inflammation did not meet the histological criteria for the diagnosis of either condition, therefore, they were identified as u-IBD. The most frequent gastrointestinal symptoms at the onset of IBD were diarrhea (17/40; 43%), followed by hematochezia or blood in stools (12/40; 30%). In 28% (11/40) of patients, the only complained symptom was unspecific abdominal pain.

In 27 cases out of 46 patients (59%) with available data, the diagnosis of CNO preceded the diagnosis of IBD, and the median time between the two events was 24 months (IQR 40). Contrariwise, in 11 patients (24%), IBD anticipated bone symptoms with a median time between the two diagnoses of 24 months (IQR 40.3). The two conditions were diagnosed simultaneously in only eight patients (17%). We divided the patients into two groups: i. patients firstly diagnosed with CNO (CNO-first) and ii. patients firstly diagnosed with IBD (IBD-first). No statistically significant differences in CD/UC (*p*-value = 0.425), constitutional symptoms (*p* = 0.625), or sex prevalence (*p*-value = 0.709) were found between the two groups.

In contrast, the use of biological disease-modifying antirheumatic drugs (bDMARDs) differed between the two groups. In the CNO-first group, 1 patient out of 27 (4%) was treated with etanercept, and the remaining received a TNFα inhibitor (TNFα-i) only after the diagnosis of IBD, while in the IBD-first group, 5 patients out of 11 (45%) were under TNFα-i treatment when bone symptoms appeared (*p*-value = 0.004; [95% CI 1.7–1054.4]). Notably, in these patients, CNO had developed despite the previous use of TNFα-i (infliximab *n* = 4, adalimumab *n* = 1) for IBD treatment with a median time between the two conditions of 36 months (IQR 36).

In one patient (Group IBD-first), increasing the dosage of infliximab led to an improvement in bone symptoms. One patient switched from infliximab to ustekinumab, an anti-IL12/23 monoclonal antibody, and the other three patients experienced an improvement in CNO while remaining on the same anti-TNFα therapy.

In the CNO-first group, the different cumulative treatments used before and after IBD diagnosis are summarized in Table 2.

No further analysis on possible predictors of developing IBD or CNO, according to the first diagnosis, was undertaken, given the low numbers and the insufficiency of the extrapolated data.

### 3.2. Comparing Literature Data with the Italian Cohort

Twenty-three patients were included in the Italian cohort. The median age at the onset of CNO was 10 years (IQR 1.5), and the median follow-up time was 90 months (IQR 86). None of the patients in our cohort developed gastrointestinal symptoms or IBD over time. Every patient was screened with fecal calprotectin at the beginning and after one year from the diagnosis, but none had a positive test. The presenting clinical manifestations were bone pain (99%), with swelling of the affected bone in 22%, and constitutional signs, including fever (17%). Nineteen patients (83%) had a polyostotic disease, while four remained monostotic (50% with sternoclavicular involvement). Clinical and demographic features of the two cohorts are compared in Table 3.

## 4. Discussion

The association between CNO and IBD was described for the first time by Kahn et al. [76] in a case series. Afterward, an increasing number of cases reporting this association was published. Although most articles concern the pediatric population, in eight reported cases the first symptoms began after 18 years of age. Although the etiology of CNO is uncertain, the strong association with other inflammatory disorders might suggest a common pathophysiology. Some interesting hypotheses have been proposed. As is the case in the pathogenesis of spondyloarthritis (SpA), it was assumed that, also in CNO patients, an impaired intestinal barrier and a consequent inflammation might release cytokines, which can mediate the extraintestinal manifestation of IBD, particularly bone inflammation. In adults affected by spondylarthritis, a subclinical gut inflammation documented by ileocolonscopy was found in around 60% of patients [77,78,79,80]. Similar results were found in patients with juvenile SpA (jSpA) [81]. In patients diagnosed with enthesitis-related arthritis (ERA), calprotectin levels were influenced by bone disease status; indeed, patients with an active disease had higher levels of fecal calprotectin compared with those with a quiescent disease [82]. Another study found an increased incidence of IBD in jSpA patients compared to the healthy population [83,84,85]. These findings suggest a potential role of dysbiosis in the pathogenesis of both diseases: an imbalance in the microbiota may lead to an increase in gut wall permeability and a defective defense mechanism permitting antigens to enter the circulation and cause inflammation elsewhere [86]. Moreover, in adult SpA, gut inflammation correlates with both disease activity and progression [79,87]. Regarding genetic factors, the best-known susceptibility genes for IBD (i.e., CARD15/NOD2 gene) were tested in a small cohort of patients with CNO; however, no association was found [63].

Fifty-seven cases of patients diagnosed with both CNO and IBD have been reported in the literature. In most of the cases (59%), bone pain and CNO diagnosis preceded gastrointestinal symptoms. However, the latency between the first and second disease was the same between CNO-first and IBD-first groups (median time of 2 years). The majority of the patients were diagnosed with CD rather than with UC (60% vs. 34%).

In the literature-review-based cohort, the most common associated disease was psoriasis, and, as occurs for IBD, primary sclerosing cholangitis was the most common hepatobiliary complication [88,89].

We found a higher incidence of constitutional signs and symptoms in our literature-review-based cohort of CNO-IBD patients compared to CNO patients in the literature (21% vs. 5%) [9,30], but no significant difference when compared with our Italian cohort (21% vs. 17%; *p*-value = 0.758). Continuing the comparison, it seems that CNO-IBD patients had a lower incidence of rachis involvement compared to CNO patients from the Italian cohort (0% vs. 26%; *p*-value = 0.001).

Another interesting observation can be made regarding the use of bDMARDs when comparing patients with a first diagnosis of CNO and patients with a first diagnosis of IBD. In the first group, only one patient was treated with bDMARDs (etanercept) before the onset of gastrointestinal symptoms, while in the second group, five patients were under TNFα-i treatment when bone symptoms appeared. In this scenario, it seems that therapy with TNFα-i does not prevent the development of bone disease. Nevertheless, the efficacy of TNFα-i y in the treatment of NSAIDs refractory CNO is well known [90], and in some patients in the literature-review-based cohort, dosage adjustment led to bone disease remission. 

The limitations of the present manuscript include the review analysis and the subsequent retrospective design. The time period of more than 20 years over which the published cases were reported may represent an undeniable bias. Furthermore, data from the literature might be incomplete and lack longitudinal details; therefore, a careful interpretation of each datum is needed. Conversely, our cohort represented a uniform sample; however its retrospective nature and the low number forbid any generalizable results. Nevertheless, this is the first complete literature review concerning the association between CNO and IBD. We not only reported published cases of this association, but we also compared this literature-review-based cohort with our cohort of CNO patients without IBD to find any possible difference.

## 5. Conclusions

Pediatric rheumatologists and gastroenterologists should be aware of the possibility of the association between the two diseases, especially in the presence of constitutional symptoms despite the ongoing treatment. The link between autoinflammatory bone disease and intestinal inflammation should be further investigated since it might offer insights into bone inflammation and new therapeutic options. We aim to increase the awareness of this association and its related characteristics in order to help clinicians in the early diagnosis and proper treatment.

## Figures and Tables

**Table 1 children-10-00502-t001:** Clinical and demographic characteristics of reviewed cases of CNO associated with IBD.

	Literature-Review-Based Cohort, Positive Findings/Pts with Available Information (%)
**Female sex**	26/47 (57)
**Polyostotic disease**	35/42 (83)
**Family history of autoimmune disease (1–2° degree)**	
Psoriasis	3/56 (5)
IBD	3/56 (5)
**Constitutional symptoms at the onset**	9/42 (21)
**Laboratory features the onset**	
ANA positivity	3/17 (18)
ANCA positivity	3/17 (18)
Low Hb levels	14/17 (83)
High calprotectin levels	10/12 (83)
High ESR/RCP	38/41 (93)

IBD = inflammatory bowel disease; ANA = antinuclear antibodies; ANCA = anti-neutrophil cytoplasmic antibodies; Hb = hemoglobin; ESR = erythrocyte sedimentation rate; CRP = reactive c protein; Pts = patients.

**Table 2 children-10-00502-t002:** Patient firstly diagnosed with CNO: treatment differences before and after IBD diagnosis.

Treatments	Before IBD Diagnosis	After IBD Diagnosis	*p*-Value
NSAIDs	15	0	0.000
Steroids	7	12	0.237
Salicylates	0	13	0.000
MTX	2	4	0.666
IVIG	1	0	1
AZA	0	6	0.021
Anti-TNFα	1	6	0.097

NSAIDs = non-steroidal anti-inflammatory drugs; MTX = methotrexate; IVIG = intravenous immunoglobulins; AZA = azathioprine; TNFα-i = tumor necrosis factor α inhibitors; IBD = inflammatory bowel diseases.

**Table 3 children-10-00502-t003:** Demographical and clinical features of CNO in the Italian cohort vs. literature-review-based cohort.

	Italian Cohort Positive Findings/Pts with Available Information (%)	Literature-Review-Based Cohort Positive Findings/Pts with Available Information (%)	*p*-Value
**Female sex**	18/23 (78)	26/47 (57)	0.071
**Age at onset, median (IQR)**	10 (1.5)	11 (4.5)	0.095
**Comorbidity**			
Psoriasis	0/23 (0)	5/56 (9)	0.314
Erythema nodosum	0/23 (0)	2/56 (4)	1
Sclerosing colanghitis	0/23 (0)	2/56 (4)	1
Hypothyroidism	2/23 (9)	0/56 (0)	0.082
Celiac disease	1/23 (4)	0/56 (0)	0.291
Takayasu vasculitis	0/23 (0)	1/56 (2)	1
Pyoderma gangrenosum	1/23	1/56 (2)	0.500
**Polyostotic disease**	19/23 (83)	35/42 (83)	1
**Constitutional symptoms at onset**	4/23 (17)	9/42 (21)	0.758
**Bone pain at the onset**			
Knee	2/23 (9)	5/42 (12)	1
Ankle	2/23 (9)	10/42 (24)	0.187
Foot	1/23 (4)	3/42 (7)	1
Jaw	0/23 (0)	3/42 (7)	0.546
Sternoclavear	3/23 (13)	6/42 (14)	1
Pelvis and SI joints	4/23 (17)	5/42 (12)	0.709
Tibia	2/23 (9)	2/42 (5)	0.610
Not specific lower extremity pain	3/23 (13)	8/42 (19)	0.733
Diffuse bone pain	3/23 (13)	3/42 (7)	0.686
Rachis	6/23 (26)	0/42 (0)	0.001
**First line therapy in patients with CNO first presentation**			
NSAIDs only	4/22 (18)	13/20 (65)	0.004
NSAIDs + csDMARDs	4/22 (18)	2/20 (10)	0.665
Bisphosphonates only	7/22 (32)	1/20 (5)	0.047
Bisphosphonates + csDMARDs	6/22 (27)	0/20 (0)	0.021
Steroids	0/22 (0)	5/20 (25)	0.018

NSAIDs = non-steroidal anti-inflammatory drugs; csDMARDs = conventional disease modifying antirheumatic drugs; CNO = chronic nonbacterial osteomyelitis.

## Data Availability

The authors confirm that the data supporting the findings of this study are available within the article.

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
