# Peer review of "Chronic Nonbacterial Osteomyelitis and Inflammatory Bowel Disease: A Literature Review-Based Cohort"

_children, 2023, doi:10.3390/children10030502_

Round 1

Reviewer 1 Report

Congratulations. The subject of this study is relevant, but the authors must improve its presentation. I am suggesting some alterations and inclusions that the authors will find in the attached file. 

Author Response

Thank you so much for the precious suggestions. We reedited the manuscript following your suggestions. We changed the title as suggested.

Reviewer 2 Report

Dear authors thank you for an interesting article but please add strengths and weaknesses to your article

Author Response

Very glad for your suggestions. We added strengths and weaknesses of our article in the last paragraph of the discussion.

Reviewer 3 Report

The authors conducted a literature review of published cases of CNO associated with IBD. The review is thorough. I have one major point to make:

1. Can the authors please explain why they compared cases of CNO associated with IBD reported in literature vs their patients who only had CNO? This needs to be clearly discussed in the manuscript.

Author Response

Thank you so much for the useful suggestion. We added an explanation regarding the comparison with our cohort: “This review aims to describe all the cases reported in the literature concerning the association between CNO and IBD and compare them with data from our single-center cohort of patients affected by CNO but without a diagnosis of IBD. The comparison was assessed to comprehend if there are clinical or demographical differences between CNO alone or in association with IBD”.

Round 2

Reviewer 1 Report

Congratulations. The manuscript was improved,

Reviewer 3 Report

The authors have revised their manuscript and adequately addressed my comments.